# EMBODIED MULTIMODAL MULTITASK LEARNING

## ABSTRACT

Visually-grounded embodied language learning models have recently shown to be effective at learning multiple multimodal tasks such as following navigational instructions and answering questions. In this paper, we address two key limitations of these models, (a) the inability to transfer the grounded knowledge across different tasks and (b) the inability to transfer to new words and concepts not seen during training using only a few examples. We propose a multitask model which facilitates knowledge transfer across tasks by disentangling the knowledge of words and visual attributes in the intermediate representations. We create scenarios and datasets to quantify cross-task knowledge transfer and show that the proposed model outperforms a range of baselines in simulated 3D environments. We also show that this disentanglement of representations makes our model modular and interpretable which allows for transfer to instructions containing new concepts.[†]

## 1 INTRODUCTION

Humans learn language by interacting with a dynamic perceptual environment, grounding words into visual entities and motor actions (Smith and Gasser, 2005; Barsalou, 2008). In recent years, there has been an increased focus on training embodied agents capable of visually-grounded language learning. These include multimodal tasks involving *one-way* communication, such as mapping navigational instructions to actions (MacMahon et al., 2006; Chen and Mooney, 2011; Artzi and Zettlemoyer, 2013; Mei et al., 2016; Misra et al., 2018); and tasks involving *two-way* communication such as embodied question answering (Gordon et al., 2018; Das et al., 2018) and embodied dialogue (de Vries et al., 2018). Other studies have shown that grounded semantic goal navigation agents can be effective at exploiting the compositionality of language to generalize to unseen instructions with an unseen composition of semantic attributes (Hermann et al., 2017; Chaplot et al., 2018), or an unseen composition of steps in a multi-step instruction (Oh et al., 2017).

However, current grounded language learning models have certain limitations. Firstly, these models are typically trained only for a single multimodal task and lack the ability to transfer grounded knowledge of 'concepts'[*] across tasks. For example, if an agent learns to follow the instruction 'Go to the red torch' and answer the question 'What color is the pillar?', then ideally it should also understand 'Go to the red pillar' and 'What color is the torch?' without additional training. Training multitask grounded-language models can also improve training sample efficiency, as these multimodal tasks share many common learning challenges including perception, grounding, and navigation.

The second limitation is the inability of trained models to quickly transfer to tasks involving unseen concepts. For example, consider a household instruction-following robot trained on an existing set of objects. We would like the robot to follow instructions involving a new object 'lamp' that has been added to the house. Existing models would need to be trained with the new object, which typically requires thousands of samples and can also lead to catastrophic forgetting of known objects. Even if the models were given some labeled samples to detect the new objects, they would require additional training to learn to combine existing grounded knowledge with the new concept (e.g., 'blue lamp' if 'blue' is already known).

In this paper, we train a multimodal multitask learning model for two tasks: *Semantic Goal Navigation*, where the agent is given a language instruction to navigate to a goal location, and *Embodied Question Answering*, where the agent is asked a question and it can navigate in the environment to gather information to answer the question (see Figure 5). We make the following contributions in this paper:

---

[†]See `https://sites.google.com/view/emml-iclr2020` for demo videos.

[*]In this paper, we refer to the knowledge of a word and its grounding in the visual world as the knowledge of a concept (for example, concept 'torch' involves word 'torch' and how torch looks visually).

Table 1: Table showing training and test sets for both the tasks, Semantic Goal Navigation (SGN) and Embodied Question Answering (EQA). The test set consists of unseen instructions and questions. The dataset evaluates a model for cross-task knowledge transfer between the embodied multimodal tasks, SGN and EQA.

| Task | Train Set | Test Set |
|------|-----------|----------|
| SGN | Instructions *not* containing 'red' & 'pillar': 'Go to the **blue** object' 'Go to the **torch**' | Instructions containing 'red' or 'pillar': 'Go to the red pillar' 'Go to the tall red object' |
| EQA | Questions *not* containing 'blue' & 'torch': 'Which object is red in color?' 'What color is the tall pillar?' | Questions containing 'blue' or 'torch': 'Which object is **blue** in color?' 'What color is the **torch**?' |

First, we define a *cross-task knowledge transfer* evaluation criterion to test the ability of multimodal multi-task models to transfer knowledge of concepts across tasks. We show that several prior single-task models, when trained on both tasks, fail to achieve cross-task knowledge transfer. This is because the visual grounding of words is often implicitly learned as a by-product of end-to-end training of the underlying task, which leads to the entanglement of knowledge of concepts in the learnt representations. We propose a novel Dual-Attention model which learns task-invariant disentangled visual and textual representations and explicitly aligns them with each other. We create datasets and simulation scenarios for testing cross-task knowledge transfer and show an absolute improvement of 43-61% on instructions and 5-26% for questions over baselines.

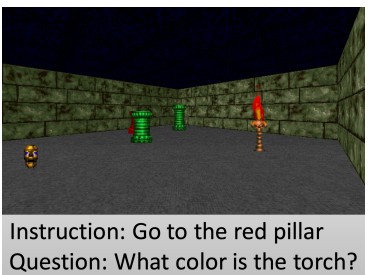

Instruction: Go to the red pillar
Question: What color is the torch?

Figure 1: Examples of embodied multimodal tasks, following instructions and answering questions.

Second, the disentanglement and explicit alignment of representations makes our model modular and interpretable. We show that this allows us to transfer the model to handle instructions involving unseen concepts by incorporating the output of object detectors. We also show that our model is able to combine the knowledge of existing concepts with a new concept without any additional policy training.

Finally, we show that the modularity and interpretability of our model also allow us to use trainable neural modules (Andreas et al., 2016) to handle relational tasks involving negation and spatial relationships and also tackle relational instructions involving new concepts.

## 2 RELATED WORK

A lot of early work on visual instruction-following in the embodied space such as in robotics applications (Tellex et al., 2011; Matuszek et al., 2012; Hemachandra et al., 2015; Misra et al., 2016) and on mapping natural language instructions to actions (MacMahon et al., 2006; Chen and Mooney, 2011; Artzi and Zettlemoyer, 2013; Mei et al., 2016) required hand-designed symbolic representations. Recently, there have been efforts on learning to follow navigational instructions from raw visual observations (Anderson et al., 2018; Misra et al., 2018; Chen et al., 2019; Blukis et al., 2018). Oh et al. (2017); Chaplot et al. (2018); Hermann et al. (2017) study the language learning aspect of instruction-following in a more controlled setting, and show that grounded language learning agents are able to learn spatial and logical reasoning and exploit the compositionality of language to generalize to new instructions.

Question Answering in the embodied space has been comparatively less-studied with recent work studying QA which requires exploration, navigation, and interaction with objects in the environment (Gordon et al., 2018; Das et al., 2018). In contrast to the prior work which tackles a single grounding task, we tackle both instruction-following and question answering in the embodied space and study the ability to transfer the knowledge of concepts across the tasks and tackle instructions with new concepts.

In addition to the above, there is a large body of work on multimodal learning in static settings which do not involve navigation or reinforcement learning. Some relevant works which use attention mechanisms similar to the ones used in our proposed model include Perez et al. (2018); Fukui et al. (2016); Xu and Saenko (2016); Hudson and Manning (2018); Gupta et al. (2017) for Visual Question Answering and Zhao et al. (2018) for grounding audio to vision.

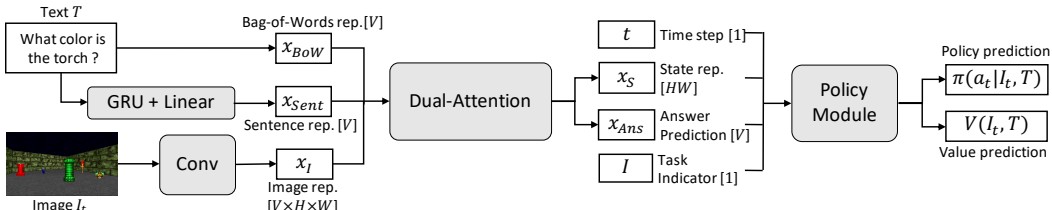

Figure 3: Overview of our proposed architecture, described in detail in Section 4.

## 3 PROBLEM FORMULATION

Consider an autonomous agent interacting with an episodic environment as shown in Figure 1. At the beginning of each episode, the agent receives a textual input $T$ specifying a task. $T$ could be an instruction to navigate to a target object or a question querying some visual detail of objects in the environment. At each time step $t$, the agent observes a state $s_t = (I_t, T)$ where $I_t$ is the first-person (egocentric) view of the environment, and takes an action $a_t$, which could be a navigational action or an answer action. The agent's objective is to learn a policy $\pi(a_t|s_t)$ which leads to successful completion of the task specified by $T$.

**Environments**. We adapt the ViZDoom-based (Kempka et al., 2016) language grounding environment proposed by Chaplot et al. (2018) for embodied multitask learning. It consists of a single room with 5 objects. The objects are randomized in each episode based on the textual input. We use two difficulty settings for the Doom domain as shown in Figure 2: *Easy*: The agent is spawned at a fixed location. The candidate objects are spawned at five fixed locations along a single horizontal line in the field of view of the agent. *Hard*: The candidate objects and the agent are spawned at random locations and the objects may or may not be in the agent's field of view in the initial configuration. The agent must explore the map to view all objects. The agent can take 4 actions: 3 navigational actions (forward, left, right) and 1 answer action. When the agent takes the answer action, the answer with the maximum probability in the output answer distribution is used.

**Datasets**. We use the set of 70 instructions from Chaplot et al. (2018) and create a dataset of 29 questions using the same set of objects and attributes. These datasets include instructions and questions about object types, colors, relative sizes (tall/short) and superlative sizes (smallest/largest). We create train-test splits for both instructions and questions datasets to explicitly test a multitask model's ability to transfer the knowledge of concepts across different tasks. Each instruction in the test set contains a word that is never seen in any instruction in the training set but is seen in some questions in the training set. Similarly, each question in the test set contains a word never seen in any training set question. Table 1 illustrates the train-test split of instructions and questions used in our experiments[†]. Note that for the EQA trainset, unseen words can be present in the answer.

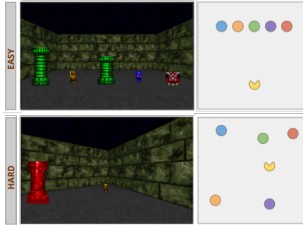

Figure 2: Sample starting states in Easy (top) and Hard (bottom) settings.

## 4 PROPOSED METHOD

In this section, we describe our proposed architecture (illustrated in Figure 3). At the start of each episode, the agent receives a textual input $T$ (an instruction or a question) specifying the task. At each time step $t$, the agent observes an egocentric image $I_t$ which is passed through a convolutional neural network (LeCun et al., 1995) with ReLU activations (Glorot et al., 2011) to produce the image representation $x_I = f(I_t; \theta_{\text{conv}}) \in \mathbb{R}^{V \times H \times W}$, where $\theta_{\text{conv}}$ denotes the parameters of the convolutional network, $V$ is the number of feature maps in the convolutional network output which is by design set equal to the vocabulary size (of the union of the instructions and questions training sets), and $H$ and $W$ are the height and width of each feature map. We use two representations for the textual input $T$: (1) the bag-of-words representation denoted by $x_{\text{BoW}} \in {0, 1}^V$ and (2) a sentence representation $x_{\text{sent}} = f(T; \theta_{\text{sent}}) \in \mathbb{R}^V$, which is computed by passing the words in $T$ through a Gated Recurrent Unit (GRU) (Cho et al., 2014) network followed by a linear layer. Here, $\theta_{\text{sent}}$ denotes the parameters of the GRU network and the linear layer with ReLU activations. Next, the

---

[†]More details about the datasets and the environments are deferred to the supplementary material.

Dual-Attention unit $f_{\text{DA}}$ combines the image representation with the text representations to get the complete state representation $x_{\text{S}}$ and answer prediction $x_{\text{Ans}}$:

$$x_{\text{S}}, x_{\text{Ans}} = f_{\text{DA}}(x_I, x_{\text{BoW}}, x_{\text{sent}}) \tag{1}$$

Finally, $x_S$ and $x_{\text{Ans}}$, along with a time step embedding and a task indicator variable (for whether the task is SGN or EQA), are passed to the policy module to produce an action.

## 4.1 DUAL-ATTENTION UNIT

The Dual-Attention unit uses two types of attention mechanisms, Gated-Attention $f_{\text{GA}}$ and Spatial-Attention $f_{\text{SA}}$, to align representations in different modalities and tasks.

**Gated-Attention (GA).** The GA unit (Chaplot et al., 2018) attends to the different channels in the image representation based on the text representation. For example, if the textual input is the instruction 'Go to the red pillar', then the GA unit can learn to attend to channels which detect red things and pillars. Specifically, the GA unit takes as input a 3-dimensional tensor image representation $y_I \in \mathbb{R}^{d \times H \times W}$ and a text representation $y_T \in \mathbb{R}^d$, and outputs a 3-dimensional tensor $z \in \mathbb{R}^{d \times H \times W}$. Note that the dimension of $y_T$ is equal to the number of feature maps and the size of the first dimension of $y_I$. In the GA unit, each element of $y_T$ is expanded to a $H \times W$ matrix, resulting in a 3-dimensional tensor $M_{y_T} \in \mathbb{R}^{d \times H \times W}$, whose $(i, j, k)^{th}$ element is given by $M_{y_T}[i, j, k] = y_T[i]$. This matrix is multiplied element-wise with the image representation: $z = f_{\text{GA}}(y_I, y_T) = M_{y_T} \odot y_I$, where $\odot$ denotes the Hadamard product (Horn, 1990).

**Spatial-Attention (SA).** We propose an SA unit which is analogous to the Gated-Attention unit except that it attends to different *pixels* in the image representation rather than the channels. For example, if the textual input is the question 'Which object is blue in color?', then we would like to spatially attend to the parts of the image which contain a blue object in order to recognize the type of the blue object. The Spatial-Attention unit takes as input a 3-dimensional tensor image representation $y_I \in \mathbb{R}^{d \times H \times W}$ and a 2-dimensional spatial attention map $y_S \in \mathbb{R}^{H \times W}$, and outputs a tensor $z \in \mathbb{R}^{d \times H \times W}$. Note that the height and width of the spatial attention map are equal to the height and width of the image representation. In the spatial-attention unit, each element of the spatial attention map is expanded to a $d$ dimensional vector. This again results in a 3-dimensional tensor $M_{y_S} \in \mathbb{R}^{d \times H \times W}$, whose $(i, j, k)^{th}$ element is given by: $M_{y_S}[i, j, k] = y_S[j, k]$. Just like in the Gated-Attention unit, this matrix is multiplied element-wise with the image representation: $z = f_{\text{SA}}(y_I, y_S) = M_{y_S} \odot y_I$.

**Dual-Attention**. We now describe the operations in the Dual-Attention unit shown in Figure 4, as well as motivate the intuitions behind each operation. Given $x_I$, $x_{\text{BoW}}$, and $x_{\text{sent}}$, the Dual-Attention unit first computes a Gated-Attention over $x_I$ using $x_{\text{BoW}}$:

$$x_{\text{GA1}} = f_{\text{GA}}(x_I, x_{\text{BoW}}) \in \mathbb{R}^{V \times H \times W} \tag{2}$$

Intuitively, this GA unit grounds each word in the vocabulary with a feature map in the image representation. A particular feature map is activated if and only if the corresponding word occurs in the textual input. Thus, the feature maps in the convolutional output learn to detect different objects and attributes, and words in the textual input specify which objects and attributes are relevant to the current task. The Gated-Attention using BoW representation attends to feature maps detecting corresponding objects and attributes, and masks all other feature maps. We use the BoW representation for the first GA unit as it explicitly aligns the words in textual input irrespective of whether it is a question or an instruction.

Next, the output of the GA unit $x_{\text{GA1}}$ is converted to a spatial attention map by summing over all channels followed by a softmax over $H \times W$ elements:

$$x_{\text{spat}} = \sigma\left(\sum_i^V x_{\text{GA1}}[i, :, :]\right) \in \mathbb{R}^{H \times W} \tag{3}$$

where the softmax $\sigma(z)_j = \exp(z_j) / \sum_k \exp(z_k)$ ensures that the attention map is spatially normalized. Summation of $x_{\text{GA1}}$ along the depth dimension gives a spatial attention map which has high activations at spatial locations where relevant objects or attributes are detected. ReLU activations in the convolutional feature maps makes all elements positive, ensuring that the summation aggregates the activations of relevant feature maps.

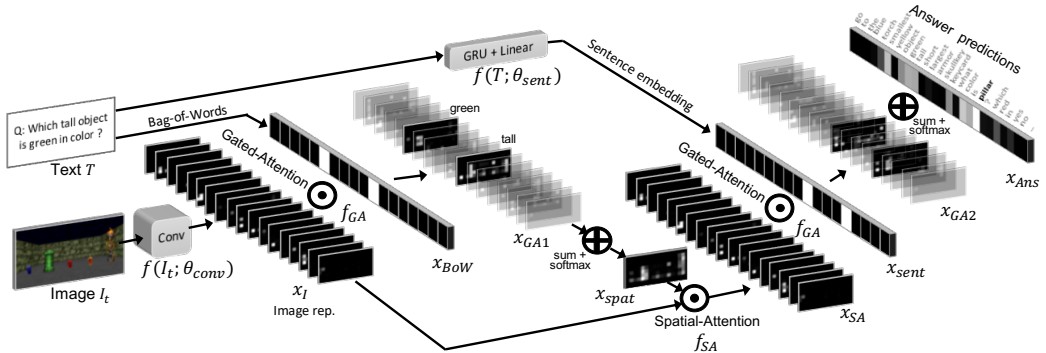

Figure 4: Architecture of the **Dual-Attention** unit with example intermediate representations and operations.

$x_{\text{spat}}$ and $x_I$ are then passed through a SA unit:

$$x_{\text{SA}} = f_{\text{SA}}(x_I, x_{\text{spat}}) \in \mathbb{R}^{V \times H \times W} \tag{4}$$

The SA unit outputs all attributes present at the locations where relevant objects and attributes are detected. This is especially helpful for question answering, where a single Gated-Attention may not be sufficient. For example, if the textual input is 'Which color is the pillar?', then the model needs to attend not only to feature maps detecting pillars (done by the Gated-Attention), but also to other attributes at the spatial locations where pillars are seen in order to predict their color.

$x_{\text{SA}}$ is then passed through another GA unit with the sentence-level text representation:

$$x_{\text{GA2}} = f_{\text{GA}}(x_{\text{SA}}, x_{\text{sent}}) \in \mathbb{R}^{V \times H \times W} \tag{5}$$

This second GA unit enables the model to attend to different types of attributes based on the question. For instance, if the question is asking about the color ('Which color is the pillar?'), then the model needs to attend to the feature maps corresponding to colors; or if the question is asking about the object type ('Which object is green in color?'), then the model needs to attend to the feature maps corresponding to object types. The sentence embedding $x_{\text{sent}}$ can learn to attend to multiple channels based on the textual input and mask the rest.

Next, the output is transformed to answer prediction by again doing a summation and softmax but this time summing over the height and width instead of the channels:

$$x_{\text{Ans}} = \sigma \left( \sum_{j,k}^{H,W} x_{\text{GA2}}[:, j, k] \right) \in \mathbb{R}^V \tag{6}$$

Summation of $x_{\text{GA2}}$ along each feature map aggregates the activations for relevant attributes spatially. Again, ReLU activations for sentence embedding ensure aggregation of activations for each attribute or word. The answer space is identical to the textual input space $\mathbb{R}^V$.

Finally, the Dual-Attention unit $f_{\text{DA}}$ outputs the answer prediction $x_{\text{Ans}}$ and the flattened spatial attention map $x_{\text{S}} = \text{vec}(x_{\text{spat}})$, where $\text{vec}(\cdot)$ denotes the flattening operation.

**Policy Module**. The policy module takes as input the state representation $x_{\text{S}}$ from the Dual-Attention unit, a time step embedding $t$, and a task indicator variable $I$ (for whether the task is SGN or EQA). The inputs are concatenated then passed through a linear layer, then a recurrent GRU layer, then linear layers to estimate the policy function $\pi(a_t \mid I_t, T)$ and the value function $V(I_t, T)$.

All above operations are differentiable, making the entire architecture trainable end-to-end. Note that all attention mechanisms in the Dual-Attention unit only modulate the input image representation, i.e., mask or amplify specific feature maps or pixels. This ensures that there is an explicit alignment between the words in the textual input, the feature maps in the image representation, and the words in the answer space. This forces the convolutional network to encode all the information required with respect to a certain word in the corresponding output channel. For example, to predict 'red' as the answer, the model must detect red objects in the corresponding feature map. This explicit task-invariant alignment between convolutional feature maps and words in the input and answer space facilitates grounding and allows for cross-task knowledge transfer. As shown in the results later, this also makes our model modular and allows easy addition of objects and attributes to a trained model.

Table 2: Accuracy of all models for both *Easy* & *Hard* difficulties. 'MT' stands for multi-task.

| | Easy | | | | | | Hard | | | | | |
| | No Aux | | | Aux | | | No Aux | | | Aux | | |
| | Train | Test | | Train | Test | | Train | Test | | Train | Test | |
| Model | MT | SGN | EQA | MT | SGN | EQA | MT | SGN | EQA | MT | SGN | EQA |
|---|---|---|---|---|---|---|---|---|---|---|---|---|
| Text only | 0.33 | 0.20 | 0.33 | 0.31 | 0.20 | 0.33 | 0.36 | 0.20 | 0.33 | 0.36 | 0.20 | 0.33 |
| Image only | 0.41 | 0.20 | 0.09 | 0.40 | 0.21 | 0.08 | 0.36 | 0.16 | 0.08 | 0.36 | 0.15 | 0.08 |
| Concat | 0.97 | 0.33 | 0.21 | **0.99** | 0.31 | 0.19 | 0.57 | 0.20 | 0.26 | 0.71 | 0.39 | 0.22 |
| GA | 0.97 | 0.27 | 0.18 | **0.99** | 0.35 | 0.24 | 0.44 | 0.18 | 0.11 | 0.71 | 0.22 | 0.24 |
| FiLM | 0.97 | 0.24 | 0.11 | **0.99** | 0.34 | 0.12 | 0.52 | 0.12 | 0.03 | 0.55 | 0.25 | 0.15 |
| PACMAN | 0.66 | 0.26 | 0.12 | 0.79 | 0.33 | 0.10 | 0.56 | 0.29 | 0.33 | 0.54 | 0.11 | 0.27 |
| Dual-Attention | **0.99** | **0.86** | **0.53** | **0.99** | **0.96** | **0.58** | **0.85** | **0.86** | **0.38** | **0.90** | **0.82** | **0.59** |

## 4.2 OPTIMIZATION

The entire model is trained to predict both navigational actions and answers jointly. The policy is trained using Proximal Policy Optimization (PPO) (Schulman et al., 2017). For training the answer predictions, we use a supervised cross-entropy loss. Both types of losses have common parameters as the answer prediction is essentially an intermediate representation for the policy.

**Auxiliary Task**. As mentioned earlier, the feature maps in the convolutional output are expected to detect different objects and attributes. Consequently, we add a spatial auxiliary task (trained with cross-entropy loss) to detect the object or attribute in the convolutional output channels corresponding to the word in the bag-of-words representation. Rather than doing fine-grained object detection, we keep the size of the auxiliary predictions the same as the convolutional output to avoid an increase in the number of parameters, and maintain the explicit alignment on the convolutional feature maps with the words. Consequently, auxiliary labels are $(V \times H \times W)$-dimensional tensors, where each of the $V$ channels corresponds to a word in the vocabulary, and each element in a channel is 1 if the corresponding object or attribute is present in the corresponding frame.

## 5 EXPERIMENTS & RESULTS

Jointly learning semantic goal navigation and embodied question answering essentially involves a fusion of textual and visual modalities. While prior methods are designed for a single task, we adapt several baselines for our environment and tasks by using their multimodal fusion techniques. We use two naive baselines, **Image only** and **Text only**; two baselines based on prior semantic goal navigation models, **Concat** (used by Hermann et al. (2017); Misra et al. (2017)) and **Gated-Attention** (GA) (Chaplot et al., 2018); and two baselines based on Question Answering models, **FiLM** (Perez et al., 2018) and **PACMAN** (Das et al., 2018). For fair comparison, we replace the proposed Dual-Attention unit with multimodal fusion techniques in the baselines and keep everything else identical to the proposed model[‡].

## 5.1 RESULTS

We train all models for 10 million frames in the *Easy* setting and 50 million frames in the *Hard* setting. We use a +1 reward for reaching the correct object in SGN episodes and predicting the correct answer in EQA episodes. We use a small negative reward of -0.001 per time step to encourage shorter paths to the target and answering questions as soon as possible. We also use distance-based reward shaping for SGN episodes, where the agent receives a small reward proportional to the decrease in distance to the target. In the next subsection, we evaluate the performance of the proposed model without the reward shaping. SGN episodes end when the agent reaches any object, and EQA episodes end when the agent predicts any answer. All episodes have a maximum length of 210 time steps. We train all models with and without the auxiliary tasks using identical reward functions.

All models are trained jointly for both the tasks and tested on each task separately[§]. In Table 2, we report the performance of all models for both *Easy* and *Hard* settings[¶]. The Dual-Attention (DA) model and many baselines achieve 99% accuracy during training in the Easy-Aux setting; however,

---

[‡]See the supplementary material for more implementation details of all baselines.

[§] See https://sites.google.com/view/emml-iclr2020/ for visualization videos.

[¶]See the supplementary material for training performance curves for all models.

Table 3: Accuracy of all the ablation models on SGN and EQA test sets for the Easy setting.

| Model | No Aux | | Aux | |
|---|---|---|---|---|
| | SGN | EQA | SGN | EQA |
| w/o SA | 0.20 | 0.16 | 0.20 | 0.15 |
| w/o GA1 | 0.14 | 0.25 | 0.16 | 0.38 |
| w/o GA2 | 0.80 | 0.33 | 0.97 | 0.15 |
| w/o Task Indicator | 0.79 | 0.47 | 0.96 | 0.56 |
| w/o Reward Shaping | 0.82 | 0.49 | 0.93 | 0.51 |
| DA Single-Task | 0.63 | 0.31 | 0.91 | 0.34 |
| DA Multi-Task | 0.86 | 0.53 | 0.96 | 0.58 |

Table 4: The performance of the trained policy appended with object detectors on instructions containing unseen words ('red' and 'pillar').

| Instruction | Acc |
|---|---|
| Go to the <color_name> **pillar**. | 1.00 |
| Go to the **red** <object_name> | 1.00 |
| Go to the tall/short **red pillar** | 0.99 |
| Go to the **red pillar** | 0.99 |
| Go to the <color_name> object that is not a **pillar** | 0.91 |
| Go to the <object_name> that is left of the **red** object | 0.96 |
| Go to the **red** object that is right of the **pillar** | 0.95 |

the test performance of all the baselines is considerably lower than that of the DA model (see Table 2 (left)). Performance of all the baselines is worse than the 'Text only' model on the EQA test set, although the training accuracy is higher. This indicates that baselines tend to overfit on the training set and fail to generalize to questions which contain words never seen in training questions. As expected, using spatial auxiliary tasks improves performance of all models. Even without auxiliary tasks, the DA model achieves a test accuracy 86% (SGN) and 53% (EQA), compared to the best baseline performance of 33% (SGN & EQA).

For the Hard setting, the DA model achieves a higher training (90% vs 71% with Aux) as well as test performance (82% vs. 39% for SGN, 59% vs. 33% for EQA with Aux) than the baselines (see Table 2 (right)). These results confirm the hypothesis that prior models, which are designed for a single task, lack the ability to align the words in both the tasks and transfer knowledge across tasks.

Lower test accuracy on EQA (vs. SGN) for most models (Table 2) indicates that EQA is more challenging as it involves alignment between not just input textual and visual representations but also with the answer space.

## 5.2 ABLATION TESTS

We perform a series of ablation tests in order to analyze the contribution of each component in the Dual-Attention unit: without Spatial-Attention (**w/o SA**), without the first Gated-Attention with $x_{BoW}$ (**w/o GA1**), and without the second Gated-Attention with $x_{\text{sent}}$ (**w/o GA2**). We also try removing the task indicator variable (**w/o Indicator Variable**), removing reward shaping (**w/o Reward Shaping**), and training the proposed model on a single task, SGN or EQA (**DA Single-Task**).

In Table 3, we report the test performance of all ablation models. The results indicate that SA and GA1 contribute the most to the performance of the full Dual-Attention model. GA2 is critical for performance on EQA but not SGN (see Table 3). This is expected as GA2 is designed to attend to different objects and attributes based on the question and is used mainly for answer prediction. It is not critical for SGN as the spatial attention map consists of locations of relevant objects, which is sufficient for navigating to the correct object [||].

We observe that reward shaping and indicator variable help with learning speed, but have little effect on the final performance (see Table 3). DA models trained only on single tasks work well on SGN, especially with auxiliary tasks, because the auxiliary task for single task models includes object detection labels corresponding to the words in the test set. This highlights a key advantage of our model's modular and interpretable design: the model can be used for transferring the policy to new objects and attributes without fine-tuning as later discussed in the Section 5.4.

## 5.3 HANDLING RELATIONAL TASKS

The instructions and questions considered so far contained a single target object. We propose a simple extension to our model to handle *relational tasks*, such as 'Which object is to the left of the torch?', where the agent is required to attend to the region *left of* the torch, not the torch itself.

---

[||] See the supplementary material for visualization of convolutional feature outputs, spatial attention map, sentence representation of the textual input and answer predictions

We consider three relational operations: 'left of', 'right of' and 'not'. We add questions and instructions with all objects and attributes using these relational operations to the existing dataset and perform experiments in the Easy-Aux setting. We assume that the knowledge of relational words, and the words they modify, are given. We train a separate module corresponding to each relational operation, and apply it to the convolutional output of the words that are modified. For example, for the above question, we apply the module for relation 'left of' to the convo-

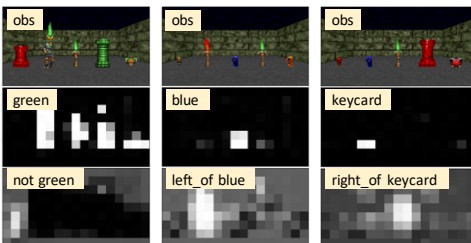

Figure 5: Outputs for relations 'not', 'left of', and 'right of' learned by the relational modules.

lutional output channel corresponding to the word 'torch'. Each relational module is a trainable convolutional network which preserves the size of the input. The rest of the operations are identical to the Dual-Attention Unit. The relational modules are learned end-to-end without any additional supervision.

In Figure 5, we show convolutional outputs of the relational modules learned by our model. While the original DA model achieves test performance of 0.48 (SGN) and 0.44 (EQA), this simple extension achieves 0.97 (SGN) and 0.64 (EQA).

## 5.4 TRANSFER TO NEW CONCEPTS

Suppose that the user wants the agent to follow instructions about a new object such as 'pillar' or a new attribute such as 'red' which the agent has never seen during training. Prior SGN models (Chaplot et al., 2018; Hermann et al., 2017; Yu et al., 2018) cannot handle instructions containing a new concept. In contrast, our model can be used for handling such instructions by training an object detector for each new concept and appending it to the image representation $x_I$. In order to test this, we train the DA model in the Easy setting on the training set for only instructions. We use auxiliary tasks but only for words in the vocabulary of the instructions training set. After training the policy, we test the agents on instructions containing test concept words 'red' and 'pillar', which the agent has never seen in textual input during training and never received any supervision about how this attribute or object looks visually.

For transferring the policy, we assume access to two object detectors for 'red' and 'pillar' separately. We resize the object detections to the size of a feature map in the image representation ($H \times W$) and append them as channels to the image representation. We also append the words 'red' and 'pillar' to the bag-of-words representations in the same order such that they are aligned with the appended feature maps. We randomly initialize the embeddings of the new words for computing the sentence embedding.

The results in Table 4 show that this policy generalizes well to different types of instructions with unseen concepts, including: combining knowledge of existing attributes with a new object, or knowledge of existing objects with a new attribute; and composing a new attribute with a new object. The results shown in the lower part of Table 4 indicate that the model also generalizes well to relational instructions containing new concepts. This means that given an object detector for a new object 'pillar', the model can (without any additional training) detect and differentiate between green and blue pillars, or between tall and short pillars; and understand left of/right of pillar, or the negation of pillar. The model can also combine 'pillar' with another new attribute 'red' to detect red pillars and understand relational instructions involving both red objects and pillars. This suggests that a trained policy can be scaled to more objects provided the complexity of navigation remains consistent.

## 6 CONCLUSION

We proposed a Dual-Attention model for visually-grounded multitask learning which uses Gated- and Spatial-Attention to disentangle attributes in feature representations and align them with the answer space. We show that the proposed model is able to transfer the knowledge of concepts across tasks and outperforms the baselines on both Semantic Goal Navigation and Embodied Question Answering by a considerable margin. We showed that disentangled and interpretablew representations make our model modular and allow for easy addition of new objects or attributes to a trained model. For future work, the model can potentially be extended to transferring knowledge across different domains by using modular interpretable representations of objects which are domain-invariant.

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

## A VISUALIZATIONS OF MODEL COMPONENTS

Figure 6 shows an example indicating that the sentence-level embedding for the question attends to relevant words. It also shows a visualization of spatial-attention maps and answer predictions for each frame in an example EQA episode. The spatial attention map shows that the model attends to relevant objects and attributes based on the question. The answer predictions change as the agent views more objects. These visualizations show that the textual and visual representations are aligned with each other, as well as with the answer space, as expected. In Figure 7, we visualize the convolutional network outputs corresponding to 7 words for the same frame for both Aux and No Aux models. As expected, the Aux model predictions are very close to the auxiliary task labels. More interestingly, the convolutional outputs of the No Aux model show that words and objects/properties in the images have been properly aligned even when the model is not trained with any auxiliary task labels.

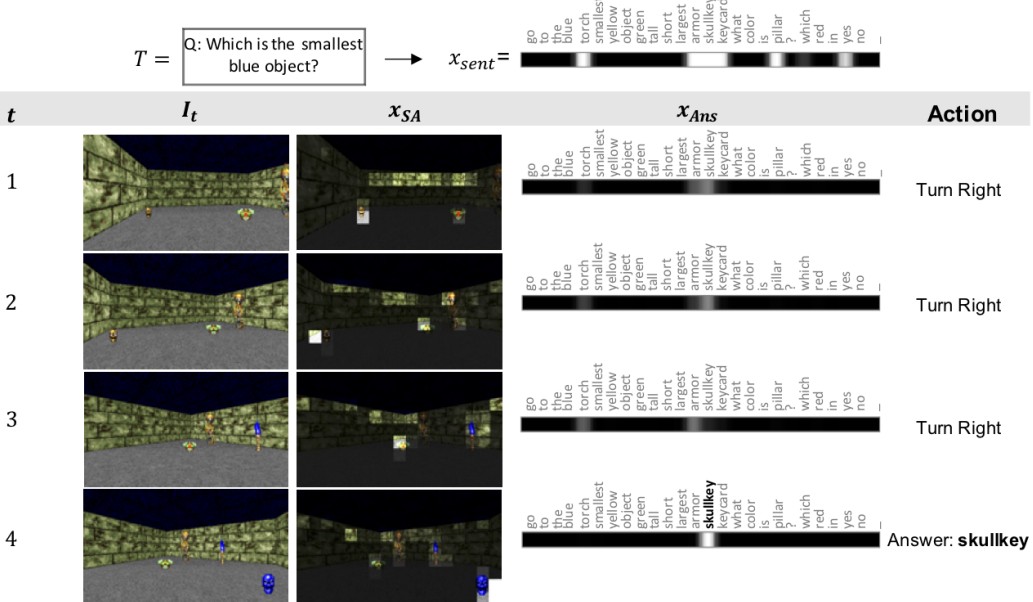

Figure 6: **Spatial Attention and Answer Prediction Visualizations.** An example EQA episode with the question "Which is the smallest blue object?". The sentence embedding of the question is shown on the top ($x_{sent}$). As expected, the embedding attends to object type words ('torch', 'pillar', 'skullkey', etc.) as the question is asking about an object type ('Which object'). The rows show increasing time steps and columns show the input frame, the input frame overlaid with the spatial attention map, the predicted answer distribution, and the action at each time step. As the agent is turning, the spatial attention attends to small and blue objects. **Time steps 1, 2**: The model is attending to the yellow skullkey but the probability of the answer is not sufficiently high, likely because the skullkey is not blue. **Time step 3**: The model cannot see the skullkey anymore so it attends to the armor which is next smallest object. Consequently, the answer prediction also predicts armor, but the policy decides not to answer due to low probability. **Time step 4**: As the agent turns more, it observes and attends to the blue skullkey. The answer prediction for 'skullkey' has high probability because it is small and blue, so the policy decides to answer the question.

Figure 7: **Visualizations of convolutional output channels.** We visualize the convolutional channels corresponding to 7 words (one in each row) for the same frame (shown in the rightmost column). The first column shows the auxiliary task labels for reference. The second column and third column show the output of the corresponding channel for the proposed Dual-Attention model trained without and with auxiliary tasks, respectively. As expected, the Aux model outputs are very close to the auxiliary task labels. The convolutional outputs of the No Aux model show that words and objects/properties in the images have been properly aligned even when the model is not trained with any auxiliary task labels. We do not provide any auxiliary label for words 'smallest' and 'largest' as they are not properties of an object and require relative comparison of objects. The visualizations in row 5 (corresponding to 'smallest') indicate that both models are able to compare the sizes of objects and detect the smallest object in the corresponding output channel even without any aux labels for the smallest object.

# B  ADDITIONAL RESULTS FOR THE DOOM ENVIRONMENT

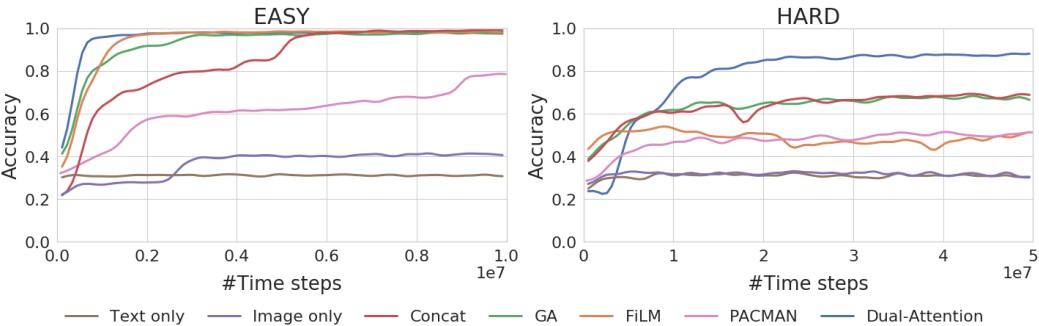

Figure 8: Training accuracy of all models trained **with** auxiliary tasks for *Easy* (left) and *Hard* (right).

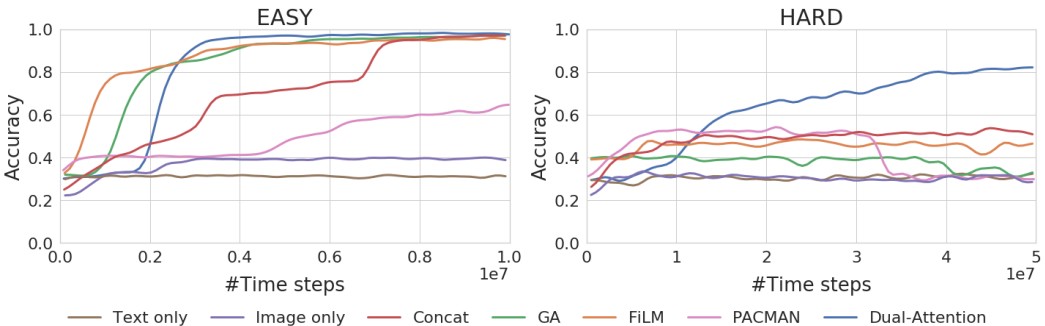

Figure 9: Training accuracy of all models trained **without** auxiliary tasks for *Easy* (left) and *Hard* (right).

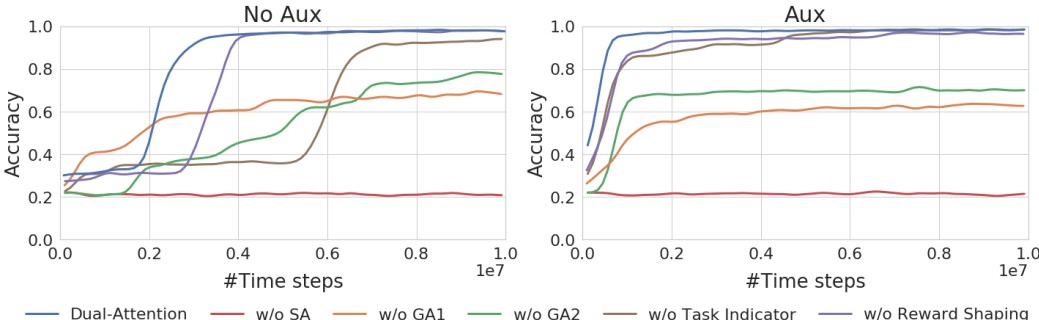

Figure 10: Training accuracy of proposed Dual-Attention model with all ablation models trained without (left) and with (right) auxiliary tasks for the *Easy* environment.

## C    DOOM ENVIRONMENT DETAILS

The Doom objects used in our experiments are illustrated in Figure 11. Instructions and questions used for training and evaluation are listed in Tables 5.

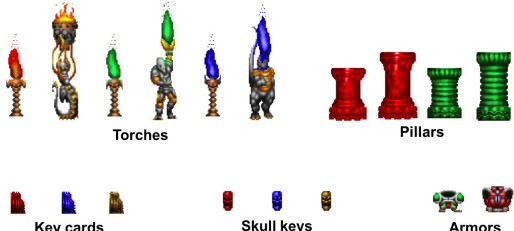

Figure 11: Objects of various colors and sizes used in the ViZDoom environment.

Table 5: Instructions and questions for ViZDoom experiments. We used 5 object classes (torch, pillar, keycard, skullkey, armor), 4 colors (red, green, blue, yellow), 2 sizes (tall, short), and 2 superlative sizes (smallest, largest).

| SGN | Instruction Type | 42 Train Instructions: Not containing 'red' & 'pillar' | 28 Test Instructions: Containing 'red' or 'pillar' |
|---|---|---|---|
| | Go to the ⟨object⟩. | torch, keycard, skullkey, armor | pillar |
| | Go to the ⟨color⟩ object. | yellow, green, blue | red |
| | Go to the ⟨size⟩ object. | tall, short | |
| | Go to the ⟨color⟩ ⟨object⟩. | blue torch, green torch, green armor, blue skullkey, blue keycard, yellow keycard, yellow skullkey | red torch, red skullkey, red pillar, green pillar, red keycard, red armor |
| | Go to the ⟨size⟩ ⟨object⟩. | short torch, tall torch | tall pillar, short pillar |
| | Go to the ⟨color⟩ ⟨size⟩ object. | green tall, blue tall, blue short, green short | red short, red tall |
| | Go to the ⟨size⟩ ⟨color⟩ object. | tall green, tall blue, short blue, short green | short red, tall red |
| | Go to the ⟨color⟩ ⟨size⟩ ⟨object⟩. | green tall torch, green short torch, blue short torch, blue tall torch | red short pillar, red short torch, red tall pillar, green tall pillar, red tall torch, green short pillar |
| | Go to the ⟨size⟩ ⟨color⟩ ⟨object⟩. | tall green torch, short green torch, short blue torch, tall blue torch | short red pillar, short red torch, tall red pillar, tall green pillar, tall red torch, short green pillar |
| | Go to the ⟨superlative⟩ object. | largest, smallest | |
| | Go to the ⟨superlative⟩ ⟨color⟩ object. | smallest yellow, smallest blue, smallest green, largest blue, largest green, largest yellow | largest red, smallest red |
| EQA | Question Type | 21 Train Questions: Not containing 'blue' & 'torch' | 8 Test Questions: Containing 'blue' or 'torch' |
| | What color is the ⟨object⟩? | pillar, keycard, skullkey, armor | torch |
| | What color is the ⟨size⟩ ⟨object⟩? | short pillar, tall pillar | short torch, tall torch |
| | Which object is ⟨color⟩ in color? | red, yellow, green | blue |
| | Which ⟨size⟩ object is ⟨color⟩ in color? | short red, tall red, short green, tall green | short blue, tall blue |
| | Which is the ⟨superlative⟩ object? | largest, smallest | |
| | Which is the ⟨superlative⟩ ⟨color⟩ object? | largest red, largest yellow, largest green, smallest red, smallest yellow, smallest green | largest blue, smallest blue |

# D  ADDITIONAL EXPERIMENTAL DETAILS

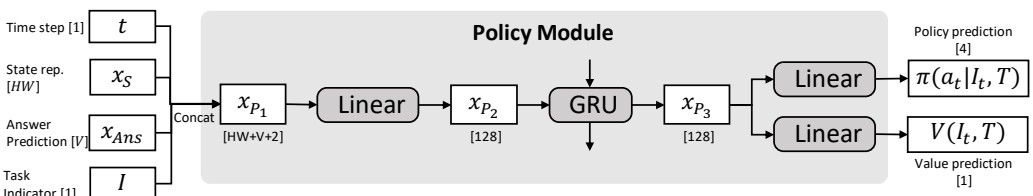

Figure 12: Architecture of the policy module.

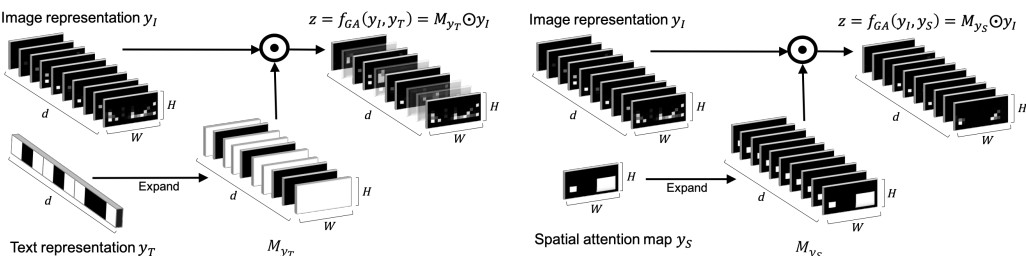

Figure 13: Gated-Attention unit $f_{\text{GA}}$

Figure 14: Spatial-Attention unit $f_{\text{SA}}$

## D.1  HYPERPARAMETERS AND NETWORK DETAILS

The input image is rescaled to size $3 \times 168 \times 300$. The convolutional network for processing the image consisted of 3 convolutional layers: conv1 containing 32 8x8 filters with stride 4, conv2 containing 64 4x4 filters with stride 2, and conv3 containing $V$ $3 \times 3$ filters with stride 2. We use ReLU activations for conv1 and conv2 and sigmoid for conv3, as its output is used as auxiliary task predictions directly. We use word embeddings and GRU of size 32 followed by a linear layer of size $V$ to get the sentence-level representation. The policy module uses hidden dimension 128 for the linear and GRU layers (see Figure 12).

For reinforcement learning, we use Proximal Policy Optimization (PPO) with 8 actors and a time horizon of 128 steps. We use a single batch with 4 PPO epochs. The clipping parameter for PPO is set to 0.2. The discount factor ($\gamma$) is 0.99. We used Adam optimizer with learning rate 2.5e-4 for all experiments.

Figures 13 and 14 illustrate the Gated-Attention unit and Spatial-Attention unit discussed in Section 4.1.

## D.2  RELATIONAL TASKS

Here, we discuss additional experimental details for the relational tasks described in Section 5.3. We consider three relations, $\mathcal{R} = \{\text{'not'}, \text{'left of'}, \text{'right of'}\}$. As mentioned previously, we assume that the knowledge of relational words, and which words are modified by relational words, are given as input to the model.

### D.2.1  EXTENDED DUAL-ATTENTION ARCHITECTURE

Given textual input $T$ and a relation $r \in \mathcal{R}$, define $y_r(T) \in \mathbb{R}^V$ to be the indicator vector for words that are modified by relation $r$. For example, if $T$ is the instruction 'Go to the torch that is not red and left_of pillar', then $y_{\text{not}}(T)$ has 1 at the index for 'red' and 0 everywhere else, and $y_{\text{left\_of}}(T)$ has 1 at the index for 'pillar' and 0 everywhere else.

Given an indicator vector $y \in \mathbb{R}^V$, let $M_y \in \mathbb{R}^{V \times H \times W}$ be the 3-dimensional tensor obtained by expanding each element of $y$ to a $H \times W$ matrix. Thus, the $(i, j, k)$th element is given by $M_y[i, j, k] = y[i]$. This matrix is multiplied element-wise with a convolutional output tensor $x \in \mathbb{R}^{V \times H \times W}$ to select the channels corresponding to words indicated by $y$.

In the extended Dual-Attention architecture, we train a separate module $f_r$ for each relation $r \in \mathcal{R}$, where each $f_r$ is a convolutional network ($5 \times 5$ kernel size, stride 1, and padding 2) that preserves the size of the input. We apply the module $f_r$ to the channels in $x_{\text{GA1}} \in \mathbb{R}^{V \times H \times W}$ which correspond to the words that are modified by relation $r$:

$$x_{\text{RA}} = \left( \mathbf{1}_{V \times H \times W} - \sum_{r \in \mathcal{R}} M_{y_r(T)} \right) \odot x_{\text{GA1}} + \sum_{r \in \mathcal{R}} M_{y_r(T)} \odot f_{\text{RA}}(x_{\text{GA1}}) \in \mathbb{R}^{V \times H \times W} \quad (7)$$

Then we use $x_{\text{RA}}$ to create the spatial attention map in equation 3:

$$x_{\text{spat}} = \sigma \left( \sum_i^V x_{\text{RA}}[i, :, :] \right) \in \mathbb{R}^{H \times W} \quad (8)$$

We also zero out the convolutional channels of $x_I$ corresponding to relational words, in order to improve generalization across different modified words. The rest of the operations are identical to the Dual-Attention architecture.

### D.2.2    DATASET GENERATION

The relational tasks are generated in the Easy-Aux setting (Section 5.3), which has five candidate objects spawned along a horizontal line in the field of view of the agent. At the start of each episode, we sample a word from $\mathcal{R} \cup \{\text{'none'}\}$.

- If 'none' is sampled, then an instruction (SGN) or a question (EQA) is sampled identically as in the original experiments.
- If 'not' is sampled, then we sample an instruction (SGN) or a question (EQA) from the original dataset, then sample a word in this instruction or question and negate it (e.g. 'Go to the red torch' $\rightarrow$ 'Go to the torch which is not red').
- If 'left of' is sampled, then we sample a correct object among the candidate objects in positions $\{0, 1, 2, 3\}$, generate a short description $x$ for the object immediately to the right of the correct object (e.g., 'red object', ',red torch', 'short torch', or 'torch'), and append 'left of $x$' to the instruction or question.
- If 'right of' is sampled, then we sample a correct object among the candidate objects in positions $\{1, 2, 3, 4\}$, generate a short description $x$ for the object immediately to the left of the correct object, and append 'right of $x$' to the instruction or question.

### D.3    BASELINE DETAILS

**Image only**: Naive baseline of just using the image representation: $x_{\text{S}} = \text{vec}(x_I)$ where $\text{vec}(.)$ denotes the flattening operation.

**Text only**: Naive baseline of just using the textual representations: $x_{\text{S}} = [x_{\text{BoW}}, x_{\text{sent}}]$.

**Concat**: The image and textual representations are concatenated: $x_{\text{S}} = [\text{vec}(x_I), x_{\text{BoW}}, x_{\text{sent}}]$. Note that concatenation is the most common method of combining representations. Hermann et al. (2017) concatenate convolutional image and bag-of-words textual representations for SGN, whereas Misra et al. (2017) use concatenation with sentence-level textual representations.

**Gated-Attention**: Adapted from Chaplot et al. (2018), who used Gated-Attention with sentence-level textual representations for SGN: $x_{\text{S}} = f_{\text{GA}}(x_I, x_{\text{sent}})$.

**FiLM**: Perez et al. (2018) introduced a general-purpose conditioning method called Feature-wise Linear Modulation (FiLM) for Visual Question Answering. Using FiLM, $x_{\text{S}} = \gamma(x_{\text{sent}}) \odot x_I + \beta(x_{\text{sent}})$ where $\gamma(x_{\text{sent}})$ and $\beta(x_{\text{sent}})$ are learnable projections of the sentence representation.

**PACMAN**: Das et al. (2018) presented a hierarchical RL model for EQA. We adapt their method by using the attention mechanism in their QA module, which takes the last 5 frames and the text as input, and computes the similarity of the text with each frame using dot products between image and sentence-level text representations. These similarities are converted into attention weights using softmax, and the attention-weighted image features are concatenated with question embedding and

passed through a softmax classifier to predict the answer distribution. For this particular baseline, we use the last 5 frames as input at each time step, unlike the proposed model and all other baselines which use a single frame as input. The attention-weighted image features are used as the state representation. The PACMAN model used a pretrained QA module, but we train this module jointly with the Navigation model for fair comparison with the proposed model.

For each of the above method except PACMAN, we use a linear layer $f$ with ReLU activations followed by softmax $\sigma$ to get a $V$-dimensional answer prediction from the state representations: $x_{\text{Ans}} = \sigma(f(x_{\text{S}}; \theta_{Lin}))$. $x_{\text{S}}$ and $x_{Ans}$ are concatenated and passed to the policy module along with the time step and task indicator variable just as in the proposed model.

# E SCALABILITY EXPERIMENTS

We perform additional experiments in a symbolic 2D environment to test the scalability of our model with respect to vocabulary size. We use 5 different abstract attribute types (e.g., object type, size, color, texture, etc.), where each attribute can take on one of $K$ values (e.g., the 'color' attribute can take on values 'blue', 'red', etc.). We construct a square maze of size 7x7, and spawn the agent with a 3x5 field of view and 5 candidate objects at random locations identical to

Table 6: Performance of the DA model trained in a 2D environment with different vocabulary sizes.

| Vocabulary size | Number of Instructions | Number of Questions | Train MT | Test SGN | EQA |
|---|---|---|---|---|---|
| 25 | $> 5^5$ | $> 5^4$ | 0.79 | 0.77 | 0.69 |
| 100 | $> 10^6$ | $> 10^4$ | 0.77 | 0.76 | 0.64 |
| 500 | $> 10^{10}$ | $> 10^8$ | 0.73 | 0.70 | 0.60 |

the Hard setting in the 3D environment. The questions and instructions are created identically to the 3D datasets except superlative questions and instructions. We assume perfect perception, meaning that the dual-attention unit receives convolutional output which detects each attribute perfectly. We perform experiments with different values of $K \in \{5, 20, 100\}$ leading to a vocabulary size of up to 500 words and more than $10^{10}$ instructions and $10^8$ questions.

The results in Table 6 indicate that the cross-task knowledge transfer performance scales well with vocabulary size. Furthermore, results in the previous subsection indicate that attributes can be swapped in and out as per requirement due to the modularity and interpretability of the model.

## F HOUSE3D EXPERIMENTS

In the House3D domain, we train on one house environment and randomize the colors of each object at the start of each episode. The agent's spawn location is fixed. We create instructions and questions dataset for this house similar to the Doom domain. The House3D objects used in our experiments are illustrated in Figure 15. Instructions and questions used for training and evaluation are listed in Table 8.

Each model is trained for 50 million frames jointly on both SGN and EQA, without the auxiliary tasks and using identical reward functions. Similar to Doom, we use a +1 reward for reaching the correct object in SGN episodes and predicting the correct answer in the EQA episodes. We use a small negative reward of -0.001 per time step to encourage shorter paths to target and answering the questions as soon as possible. We also use distance-based reward shaping for both SGN and EQA episodes, where the agent receives a small reward proportional to the decrease in distance to the target. SGN episodes end when the agent reaches any object and EQA episodes when agent predicts any answer. All episodes have a maximum length of 420 time steps.

|  | SGN | | EQA | |
| Model | Train | Test | Train | Test |
|---|---|---|---|---|
| Text only | 0.63 | 0.33 | 0.22 | 0.23 |
| Image only | 0.28 | 0.01 | 0.12 | 0.22 |
| Concat | 0.65 | 0.13 | 0.31 | 0.13 |
| GA | 0.98 | 0.20 | **0.92** | 0.03 |
| FiLM | 0.99 | 0.37 | **0.92** | 0.24 |
| PACMAN | 0.73 | 0.20 | 0.40 | 0.21 |
| **Dual-Attention** | **0.99** | **0.47** | 0.89 | **0.29** |

Table 7: Accuracy of all the models on the SGN and EQA train and test sets for the House3D Domain.

In Table 7, we report the train and test performance of all the models on both SGN and EQA. The results are similar as in Doom: the Dual-Attention model outperforms the baselines by a considerable margin.

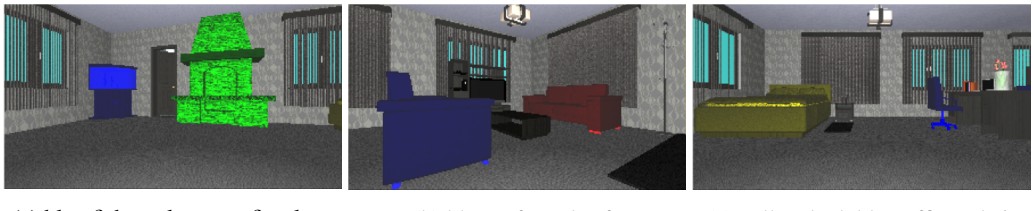

(a) blue fish_tank, green fireplace    (b) blue sofa, red sofa    (c) yellow bed, blue office_chair

Figure 15: Example first-person views of the House3D environment with sample objects of various colors.

| SGN | Instruction Type | 22 Train Instructions: 
 Not containing 'red' & 'bed' | 11 Test Instructions: 
 Containing 'red' or 'bed' |
|---|---|---|---|
| | Go to the ⟨object⟩. 
 Go to the ⟨color⟩ ⟨object⟩. | refrigerator, office_chair, fish_tank, fireplace 
 green refrigerator, green office_chair, 
 green fish_tank, green fireplace, green sofa, 
 blue refrigerator, blue office_chair, 
 blue fish_tank, blue fireplace, blue sofa, 
 yellow refrigerator, yellow office_chair, 
 yellow fish_tank, yellow fireplace, yellow sofa | bed 
 red bed, green bed, blue bed, 
 yellow bed, red refrigerator, 
 red office_chair, red fish_tank, 
 red fireplace, red sofa |
| | Go to the ⟨color⟩ object. | green, blue, yellow | red |
| EQA | Question Type | 7 Train Questions: 
 Not containing 'blue' & 'sofa' | 2 Test Questions: 
 Containing 'blue' or 'sofa' |
| | What color is the ⟨object⟩? 
 What object is ⟨color⟩ in color? | refrigerator, office_chair, fish_tank, fireplace, bed 
 red, green, yellow | sofa 
 blue |

Table 8: Instructions and questions for House3D experiments. We used 6 object classes (refrigerator, office_chair, fish_tank, fireplace, bed, sofa) and 4 colors (red, green, blue, yellow).

