# OpenReview forum: "Embodied Multimodal Multitask Learning"
_ICLR.cc/2020/Conference — Reject_

### Official Review · AnonReviewer1 · 2019-10-22
**Official Blind Review #1**

**Rating:** 6

**Review:**


The paper explores multi-task learning in embodied environments and proposes a Dual-Attention Model that disentangles the knowledge of words and visual attributes in the intermediate representations. It addresses two tasks, namely Semantic Goal Navigation (SGN) and Embodied Question Answering (EQA), using a simple synthetic environment. The paper compares against a few simple baselines and baselines adapted from models in each task.

I would recommend for acceptance, as the experimental results show that the proposed approach successfully transfers knowledge across tasks.

However, I would also like to note that the paper has a few drawbacks.

First, the paper uses a new environment to evaluate the SGN and EQA task instead of the benchmark environments for these two tasks, making it difficult to compare performance to previous work. The environment in the paper is small (compared to e.g., House3D for EQA) and has a limited variety. Also, the paper only compares to relatively out-of-date approaches on EQA and SGN, instead of the state-of-the-art approaches on them.

In addition, the paper should also discuss its connections to other multi-task learning approaches in the related work section.

**Experience Assessment:**

I have published one or two papers in this area.

**Review Assessment: Checking Correctness Of Derivations And Theory:**

N/A

**Review Assessment: Checking Correctness Of Experiments:**

I carefully checked the experiments.

**Review Assessment: Thoroughness In Paper Reading:**

I read the paper at least twice and used my best judgement in assessing the paper.

---

> ### Author Response · Authors · 2019-11-14
> **Response to Reviewer #1**
>
> Thanks for the review and helpful feedback. We address your concerns below:
>
> > First, the paper uses a new environment to evaluate the SGN and EQA task instead of the benchmark environments for these two tasks, making it difficult to compare performance to previous work
>
> We agree with you that reproducibility and benchmarking is important. And this is the reason we did not use the House3D EQA dataset as it requires the SUNCG dataset which is no longer available. For SGN, we use the same dataset as used by Chaplot et. al. 2018 [1].
>
>
> > Also, the paper only compares to relatively out-of-date approaches on EQA and SGN, instead of the state-of-the-art approaches on them.
>
> Please note that our baselines aren't weak -- as shown by the multi-task training performance of our baselines in Table 2, they achieve nearly 100% performance on both SGN and EQA during training. Testing a newer method will not improve this performance. The problem is not that these baselines are ineffective at SGN or EQA but that these models are designed for a single task and hence do not perform well when tested for cross-task knowledge transfer. And this issue remains with the state-of-the-art single-task approaches for EQA and SGN. In fact, no prior work has proposed a model for cross-task knowledge transfer for embodied multimodal learning, so we cannot easily compare to prior work (we do construct reasonable baselines and ablations, as described in section 5.1 and 5.2). Having said that, if there are any specific recommendations for a baseline to add that we may have missed, we will be happy to add it in the revised version.
>
>
> > In addition, the paper should also discuss its connections to other multi-task learning approaches in the related work section.
>
> We did not discuss multitask learning in the related work as we are not aware of any multitask learning approaches specific to embodied multimodal learning. We will add a discussion about multitask learning in non-embodied multimodal settings in the revised version.
>
> [1] Gated-Attention Architectures for Task-Oriented Language Grounding
> Devendra Singh Chaplot, Kanthashree Mysore Sathyendra, Rama Kumar Pasumarthi, Dheeraj Rajagopal, Ruslan Salakhutdinov

---

### Official Review · AnonReviewer2 · 2019-10-23
**Official Blind Review #2**

**Rating:** 3

**Review:**

*Summary

The paper describes a Dual-Attention model using Gated- and Spatial-Attention for disentanglement of attributes in feature representations for visually-grounded multitask learning. It has been shown that these models are capable of learning navigational instructions and answering questions. However, they addressed two limitations of previous works about visually-grounded embodied language learning models.  The first is the inability to transfer grounded knowledge across different
tasks, and the other is the inability to transfer to new words and concepts not seen during the training phase. To overcome the problem, a multitask model is introduced. The model can transfer knowledge across tasks via learning disentanglement of the knowledge of words and visual attributes. The paper shows that the proposed model outperforms a range of baselines in simulated 3D environments.


*Decision and supporting arguments

I think the paper is on the borderline. The reason is as follows.
The motivation of the study is described appropriately, and the performance is quantitatively evaluated, as shown while Table 2.
However, the generality of the proposed method, i.e., dual attention, is still ambiguous. Though the devised module performs effectively in this specific simulation environment and specific two tasks, an explanation of the theoretical basis and generality of dual attention seem to be missing.
Even though the title has the phrase "multitask learning," what the system copes with is just two specific tasks.  If the system is designed to solve the two specific tasks simultaneously, it's better to change the title. The title seems to be misleading.
Some of the main contributions, e.g., "modularity and interpretability" and "transfer to new concepts," are not evaluated quantitatively.


*Additional feedback
In conclusion, "interpretablew" -> "interpretable"

**Experience Assessment:**

I have read many papers in this area.

**Review Assessment: Checking Correctness Of Derivations And Theory:**

N/A

**Review Assessment: Checking Correctness Of Experiments:**

I assessed the sensibility of the experiments.

**Review Assessment: Thoroughness In Paper Reading:**

I read the paper at least twice and used my best judgement in assessing the paper.

---

> ### Author Response · Authors · 2019-11-14
> **Response to Reviewer #2**
>
> Thanks for the helpful review and feedback. We address your concerns below:
>
> > However, the generality of the proposed method, i.e., dual attention, is still ambiguous …
>
> We argue that the proposed dual-attention is generally applicable to any multimodal task which requires grounding words in visual concepts. It provides a general way of aligning textual and visual representations in any multimodal task such that they can be reused for other tasks. Although it is evaluated on two tasks in a specific environment, the design of the dual-attention unit itself is not specific to the environment or the task.
>
>
> > Even though the title has the phrase "multitask learning," what the system copes with is just two specific tasks. If the system is designed to solve the two specific tasks simultaneously, it's better to change the title. The title seems to be misleading.
>
> Note that we called semantic goal navigation as a single task in the paper for easier understanding, whereas prior work [1, 2, 3] has called each instruction as a different task and handling multiple instructions as multi-task learning. In our work, we not only handle multiple instructions but also multiple questions. Thus, under the notation of past work, we are solving multiple tasks. Having said that, we see your concern, and we are happy to change the title to “Embodied Multimodal Learning and Knowledge Transfer between Semantic Goal Navigation and Embodied Question Answering” or just “Embodied Multimodal Learning and Knowledge Transfer” if the reviewers and the Area Chair find this more suitable. We are also open to other suggestions from you.
>
>
> > Some of the main contributions, e.g., "modularity and interpretability" and "transfer to new concepts," are not evaluated quantitatively.
>
> We believe these contributions are evaluated quantitatively. In Section 5.4: "Transfer to new concepts"', we evaluate the model's capability of transferring to new concepts with quantitative results in Table 4. The results show that our model achieves a success rate of 0.97 on average over different types of instructions involving new object types and attributes. These results also demonstrate not only our model’s ability to handle new concepts but also to combine the knowledge of existing concepts with a new concept without any additional policy training. The above results are possible because of the modularity and interpretability of the model, as it allows us to add the output of external object detectors as intermediate representation in our model.
>
> Furthermore, in Section 5.3 “Handling relational tasks” we show that the modularity and interpretability of our model also allow us to use trainable neural modules to handle relational tasks involving negation and spatial relationships and also tackle relational instructions involving new concepts.
>
> Thanks for pointing out the typo. We will correct it in the revised version.
>
> [1] Zero-Shot Task Generalization with Multi-Task Deep Reinforcement Learning
> Junhyuk Oh, Satinder Singh, Honglak Lee, Pushmeet Kohli
>
> [2] Grounded Language Learning in a Simulated 3D World
> Karl Moritz Hermann, Felix Hill, Simon Green, Fumin Wang, Ryan Faulkner, Hubert Soyer, David Szepesvari, Wojciech Marian Czarnecki, Max Jaderberg, Denis Teplyashin, Marcus Wainwright, Chris Apps, Demis Hassabis, Phil Blunsom
>
> [3] Gated-Attention Architectures for Task-Oriented Language Grounding
> Devendra Singh Chaplot, Kanthashree Mysore Sathyendra, Rama Kumar Pasumarthi, Dheeraj Rajagopal, Ruslan Salakhutdinov

---

### Official Review · AnonReviewer3 · 2019-10-24
**Official Blind Review #3**

**Rating:** 3

**Review:**

I thank the authors for their detailed response and appreciate their hard work in bringing us this paper.

I think that my main point is that this work relies too much on the extra information/constraints in the synthetic env. E.g., 1. since the vocab size is small, thus the feature map could be designed 'equal to the vocabulary size' 2. The bag-of-words representation is effective but it is not the case for natural language. Although the authors kindly point me to some recent works on sim2real, I am still not convinced whether this proposed method could be transferred to real setups based on the referenced papers.

However, it is a personal research taste that I always take real setup into considerations, because I have worked on both synthetic and real setup (on both lang and visn sides) for years and observed a large gap. My opinion is that methods of synthetic setups are not naturally convertible to the real ones. If AC/meta-reviewer considers the ability of vision-and-language interactions could be effectively studied through this setup with synthetic language and simulated-unrealistic images, I am OK with acceptance. I have downgraded my confidence scores (but kept my overall score) for this purpose.


-----------------------------------------------------------------------------------

Pros:
(1) The proposed model makes sense to me, which tries to have two attention layers to extract the information related to the questions. It seems to have the ability to deal with "and"/"or" logical relationships as well.

(2) Fig. 4 is impressive. It is clear and well-designed.

(3) The results in Table 2 are convincing. They show that both the proposed dual-attention method and multi-task learning would contribute to the performance.

Cons:
(1) It seems that the two main contributions are related to the language. Thus the synthetic language might not be proper to study. For example, in Eqn. 2, the first GA multiplies the BOW vector with the vision feature map, which could filter out unrelated instruction. This method could not be directly transferred to a real setup where natural language and natural images are involved.

(2) The designed attention modules is lack of generalizability. It implements a two-step attention module, while the first step selects the related visual regions w.r.t the words and the second step gathers the information regarding these attended regions. However, it might not be aware of the spatial relationships and thus be limited to simple questions. For example, if the question is "What is the object on top of the apple?". To my understanding, the current module would not explicitly handle this one-hop spatial relationship.

Comments:
(1) According to Sec. 3, 70 instructions and 29 questions are involved in this task. Using GRU to encoder these questions seems to be redundant. A simple one-hot embedding for these instructions might already be enough to encode the information.

(2) I am not sure why the visual attention map x_S could be used as the state of the module.

(3) After Eqn. 3, the paper says that "ReLU activations ... make all elements positive, ensuring ...". I am confused about the intuition behind this argument because of the softmax activation. Softmax will projects 0 to 1. So the sum of the all-zero vector would still be non-zero after softmax.

Typo:
- In Sec. 4, X_{BoW} \in \{0, 1\}^V.
- In Sec. 4.1, "this matrix is multiplied ..." --> this tensor.

**Experience Assessment:**

I have published in this field for several years.

**Review Assessment: Checking Correctness Of Derivations And Theory:**

I assessed the sensibility of the derivations and theory.

**Review Assessment: Checking Correctness Of Experiments:**

I assessed the sensibility of the experiments.

**Review Assessment: Thoroughness In Paper Reading:**

I made a quick assessment of this paper.

---

> ### Author Response · Authors · 2019-11-14
> **Response to Reviewer #3**
>
> Thanks for the review and helpful feedback. We address your concerns and answer your questions below:
>
> Regarding the use of synthetic language: The focus of this submission is not tackling natural language but transferring the knowledge of grounded concepts (words grounded to their visual properties) across different embodied multimodal tasks and handling new concepts never seen during training. In our opinion, handling natural language is important but a separate problem in itself. For example, a parallel submission in ICLR (https://openreview.net/forum?id=rklraTNFwB) studies this problem specifically and shows that even embodied agents trained only with synthetic language can be transferred to natural language by using word representations learned by language models trained on large text corpora. Another work [1] shows that embodied instruction-following models can be transferred to unseen natural language synonymous words using GLoVe [2] word embeddings. These approaches could be used to transfer our model to natural language as well.
>
> In order to provide evidence for the above, we conducted additional experiments. We constructed a new test set using synonymous words given by [1] such that each question and instruction in this new test set contains at least one unseen word never seen in any task during training. In order to handle this new test set containing unseen natural language words, we map each unseen word to the closest seen word in our synthetic data using the GLoVe [2] word vector space similar to [1]. The Dual-Attention model achieved a performance of 0.81/0.48 SGN/EQA as compared to the best performance of 0.27/0.19 SGN/EQA (GA) among the baselines. Clearly, we understand that natural language has much more complexity than previously unseen synonyms, but these results and the papers referenced above indicate that models trained with synthetic language can be used with natural language as well.
>
>
> > it might not be aware of the spatial relationships and thus be limited to simple questions. For example, if the question is "What is the object on top of the apple?". To my understanding, the current module would not explicitly handle this one-hop spatial relationship.
>
> The current model can handle relational questions including one-hop spatial relationships as shown in Section 5.3: "Handling Relational Tasks". In this section, we show how a simple extension of the model can address questions and instructions containing "left of", "right of" and "not". We also show visualizations of the learnt representations in Figure 5. We do not tackle ‘top of’ specifically, but ‘left of’ and ‘right of’ are analogous to `top of’ and tackle one-hop spatial relationship as the review mentions.
>
>
> > I am not sure why the visual attention map x_S could be used as the state of the module.
>
> The visual attention map is passed to the navigation policy because the information in the visual attention map is sufficient for successful navigation. For example, for the instruction, `Go to the red torch’ if the visual attention map identifies the location of red and torch things, that information is sufficient for navigating to the red torch.
>
>
> > After Eqn. 3, the paper says that "ReLU activations ... make all elements positive, ensuring ...". I am confused about the intuition behind this argument because of the softmax activation. Softmax will projects 0 to 1. So the sum of the all-zero vector would still be non-zero after softmax.
>
> The purpose of ReLU activations is not to zero-out the prediction after softmax. In fact, ReLU activations were chosen independently of the subsequent softmax operation. The purpose of ReLU activations is to have only positive activations during summation. Positive activations ensure that they aggregate during summation. If there were negative activations, they could potentially cancel out positive activations during summation.
>
> Thanks for pointing out the typos. We will correct them in the revised version.
>
> [1] ACTRCE: Augmenting Experience via Teacher's Advice For Multi-Goal Reinforcement Learning
> Harris Chan, Yuhuai Wu, Jamie Kiros, Sanja Fidler, Jimmy Ba
>
> [2] Glove: Global vectors for word representation.
> Jeffrey Pennington, Richard Socher, and Christopher Manning

---

### Decision · Program_Chairs · 2019-12-19

**Decision:**

Reject

**Comment:**

This paper offers a new approach to cross-modal embodied learning that aims to overcome limited vocabulary and other issues.  Reviews are mixed.  I concur with the two reviewers who say the work is interesting but the contribution is not sufficiently clear for acceptance at this time.